# Sex Differences in Heart Failure Mortality with Preserved, Mildly Reduced and Reduced Ejection Fraction: A Retrospective, Single-Center, Large-Cohort Study

**DOI:** 10.3390/ijerph192316171

**Published:** 2022-12-02

**Authors:** Antonio de Padua Mansur, Carlo Henrique Del Carlo, Gustavo Henrique Ferreira Gonçalinho, Solange Desirée Avakian, Lucas Carrara Ribeiro, Barbara Maria Ianni, Fábio Fernandes, Luiz Antonio Machado César, Edimar Alcides Bocchi, Antonio Carlos Pereira-Barretto

**Affiliations:** 1Serviço de Prevencao, Cardiopatia na Mulher e Reabilitação Cardiovascular, Instituto do Coracao (InCor), Hospital das Clinicas HCFMUSP, Faculdade de Medicina, Universidade de Sao Paulo, Sao Paulo 05403-900, Brazil; 2Hospital Dia, Instituto do Coracao (InCor), Hospital das Clinicas HCFMUSP, Faculdade de Medicina, Universidade de Sao Paulo, Sao Paulo 05403-900, Brazil; 3Unidade Clínica de Valvopatias, Instituto do Coracao (InCor), Hospital das Clinicas HCFMUSP, Faculdade de Medicina, Universidade de Sao Paulo, Sao Paulo 05403-900, Brazil; 4Faculdade de Medicina FMUSP, Universidade de Sao Paulo, Sao Paulo 01246-903, Brazil; 5Unidade Clínica de Miocardiopatias e Doenças da Aorta, Instituto do Coracao (InCor), Hospital das Clinicas HCFMUSP, Faculdade de Medicina, Universidade de Sao Paulo, Sao Paulo 05403-900, Brazil; 6Unidade Clinica de Coronariopatias Cronicas, Instituto do Coracao (InCor), Hospital das Clinicas HCFMUSP, Faculdade de Medicina, Universidade de Sao Paulo, Sao Paulo 05403-900, Brazil; 7Unidade Clinica de Insuficiencia Cardiaca, Instituto do Coracao (InCor), Hospital das Clinicas HCFMUSP, Faculdade de Medicina, Universidade de Sao Paulo, Sao Paulo 05403-900, Brazil

**Keywords:** heart failure, cardiomyopathy, left ventricular ejection fraction, prognosis, women, men

## Abstract

Background: Heart failure (HF) is one of the leading causes of death worldwide. Studies show that women have better survival rates than men despite higher hospitalizations. However, little is known about differences in mortality and predictors of death in women and men with HF with preserved (HFpEF), mildly reduced (HFmrEF), and reduced ejection fraction (HFrEF). Methods: From February 2017 to September 2020, mortality and predictors of death were analyzed in women and men with HF. Baseline data included clinical characteristics and echocardiographic findings. Results: A total of 11,282 patients, 63.9 ± 14.4 years, including 6256 (55.4%) males, were studied. Females were older, had a higher baseline mean left ventricular ejection fraction (LVEF) and lower left ventricular diastolic diameter. During follow-ups, 1375 (22%) men and 925 (18.4%) women died. Cumulative incidence of death was higher in men with HFrEF but similar for HFmrEF and HFpEF. Cox regression for death showed renal dysfunction, stroke, diabetes, atrial fibrillation, age, LVEF, valve disease, MI, and hypertensive CMP as independent death predictors for all HF patients. Conclusions: Women had a better prognosis than men in HFrEF and similar mortality for HFmrEF and HFpEF, but sex was not an independent predictor of death for all HF subtypes.

## 1. Introduction

Heart failure (HF) is one of the leading causes of death and hospitalization from cardiovascular disease worldwide and years lived with disability [1]. In Brazil, despite the decline in the mortality rate, HF is still the leading cause of hospitalization and hospital death [2]. The HF incidence increases markedly after 60 years but more significantly in men [3]. The lifetime risk of developing HF is much higher in men, probably due to the higher prevalence of comorbidities such as myocardial infarction (MI), hypertension, diabetes, and chronic kidney disease (CKD) [4]. Women with HF are more likely to be older, hypertensive, with preserved systolic function, and have a lower incidence of coronary artery disease [5]. HF in women presents more frequently with preserved left ventricular ejection fraction (HFpEF) rather than with reduced ejection fraction (HFrEF), which occurs predominantly in men [6,7]. Myocardial dysfunction in HF activates sympathetic and renin-angiotensin-aldosterone systems, increasing LV end-diastolic pressure and afterload in women and men. Nevertheless, there are sex differences in HF pathophysiology, which depend on estrogen, gene expression, inflammation, and comorbidities. In women, these differences may be related to endogenous estrogen-mediated mechanisms [8]. Estrogen protects the epicardial and microvasculature endothelial response to injury and reduces the activation of the sympathetic and renin-angiotensin-aldosterone systems. Estrogen’s beneficial effects protect the left ventricle from remodeling and maintain diastolic function. Low estrogen levels activate these systems that favor the rise of reactive oxygen species and collagen synthesis, drops in nitric oxide, and diastolic dysfunction [9]. HF diagnosis requires typical signs and symptoms associated with increased BNP plasma concentration and a non-invasive cardiac imaging test, most often by a transthoracic echocardiogram showing left ventricular dysfunction [10]. There is evidence of a gender-associated risk of dying of HF in different HF subtypes [11,12]. The lifetime risk also differs in the HF spectrum, with HFrEF higher in men but similar in women and men for HFpEF [13]. Studies show that women have better survival rates than men for all HF subtypes, despite higher hospitalizations [14,15]. Men with HFrEF have higher cardiovascular mortality than women [16]. In HF with mid-range ejection fraction (HFmrEF) and HFpEF, women still show better survival differences than HFrEF. Nevertheless, sex-specific data are scarce concerning the epidemiology, comorbidities, hospitalizations, prognosis, and treatment of the different HF subtypes in populations of women and men. This study analyzed sex differences in all causes of death in HF subtypes in women and men.

## 2. Materials and Methods

This retrospective and single-center large cohort study analyzed clinical characteristics, laboratory data, and prognosis of 11,282 outpatients with HF aged ≥18 years, including 6256 (55.4%) men. This study was performed at the Instituto do Coração (InCor), Hospital das Clínicas, Faculdade de Medicina da Universidade de São Paulo, from February 2017 to September 2020. The Hospital Medical Information Unit provided the patients’ data. The patients of interest were those diagnosed with HF. The Framingham criteria for diagnosing HF [17] and echocardiographic measurements were used. The echocardiographic data were those of the patients with both the baseline and the closest echocardiogram available at the end of the study. HF was categorized according to LVEF in HFrEF for LVEF ≤ 40%, HFmrEF for LVEF of 41–49%, and HFpEF for LVEF ≥ 50%. We excluded patients with HF from congenital heart disease and Chagas disease. The primary outcome was all-cause mortality which included cardiovascular and non-cardiovascular death. Cardiovascular death included fatal myocardial infarction, stroke, or other cardiovascular deaths. Non-cardiovascular deaths included cancer and infection-related deaths. From March to September 2020, COVID-19 infection-related deaths may have occurred in the study population, but the exact number of COVID-19 deaths was unavailable. The mortality data were obtained from the patient’s medical record or the individual registration status on the Federal Revenue’s website [18]. The clinical characteristics analyzed were age, the prevalence of comorbidities, the number of HF hospitalizations, and any cardiac surgical intervention. The comorbidities analyzed were diabetes (glycemia ≥ 126 mg/dL or under hypoglycemic drug), significant CKD (creatinine ≥ 2 mg/dL), atrial fibrillation (AF), previous MI, and stroke. The cardiac surgical interventions analyzed were percutaneous coronary intervention (PCI), coronary artery bypass graft (CABG), valve repair or replacement, pacemaker implantation, cardiac resynchronization therapy (CRT), implantable cardioverter defibrillators (ICD), and heart transplants. 

### Statistical Analysis

The continuous variables were presented as mean, standard deviation, and categorical variables as frequencies and percentages. Normality was tested using the Kolmogorov–Smirnov test. The Student’s *t*-test and analysis of variance were used to compare groups with continuous variables and chi-square for categorical variables. A two-sided probability of <0.05 will consider the statistical significance. We used the Kaplan–Meier (K-M) method with Šidák multiple-comparison adjustment to analyze the cumulative incidence of all-cause death. Cox proportional hazards models were used to examine variables independently associated with all-cause death. The chi-square score of the Cox proportional hazards model defined the most robust predictors of all-cause death. The dependent variable in Cox proportional hazards model was death. Covariates were age, sex, MI, diabetes, stroke, CKD, AF, PCI, CABG, LVEF, all coronary surgical intervention (percutaneous coronary intervention (PCI) + coronary artery bypass graft (CABG)), and all devices implantation (pacemaker + cardiac resynchronization therapy + implantable cardioverter-defibrillator). We used the SAS^®^ Studio package (SAS Institute, Cary, NC, USA) for statistical analyses.

## 3. Results

Table 1 shows the clinical characteristics and echocardiographic data from all patients, in women and men.

Women were older and had a higher prevalence of hypertensive CMP and valve disease. The comorbidities were higher in men than women, as well as PCI, CABG, ICD, and transplant. Women had higher pacemaker implantation, hospitalizations, baseline, and final LVEF. The baseline and final LVDD were lower in women. In women, final LVEF and LVDD decreased compared with baseline LVEF and LVDD, respectively. In men, LVEF increased and LVDD decreased. During the follow-up period, 1375 (22%) men and 925 (18.4%) women died (*p* < 0.0001). The cumulative incidence of death was higher in men (K-M: log-rank *p* = 0.0001) (Figure 1).

Table 2 shows the clinical characteristics and echocardiographic data of the patients with HF with preserved (HFpEF), mid-range (HFmrEF), and reduced (HFrEF) left ventricular ejection fraction.

Patients with HFpEF were older and there was a greater number of women. They had a higher prevalence of valve disease, comorbidities, surgical procedures for valve disease, pacemaker implantation, and hospitalizations. On the other hand, patients with HFrEF had a higher prevalence of idiopathic CMP and higher rates of surgical procedures such as ICD and CRT implantation and transplants. In the same group comparisons, HFrEF patients had an increase in final LVEF and a decrease in LVDD. In patients with HFmrEF, final LVDD increased compared to baseline LVDD. In patients with HFpEF, final LVEF decreased, and LVDD increased compared with respective baseline values. During the follow-up period, 1054 (24.5%) patients with HFrEF, 342 (18.7) with HFmrEF, and 904 (17.6) with HFpEF died (*p* < 0.0001). The cumulative incidence of death was higher in men with HFrEF compared to HFmrEF (*p* < 0.001) but similar for HFmrEF and HFpEF in women and men (Figure 2).

Table 3 shows the clinical characteristics and echocardiographic data of women and men with HF with preserved (HFpEF), mid-range (HFmrEF), and reduced (HFrEF) left ventricular ejection fraction.

In HFrEF, men had a higher prevalence of ischemic CMP, comorbidities, CABG, and ICD surgical procedures. Men also had higher baseline and final LVDD. Women had a higher prevalence of idiopathic and hypertensive CMP and CRT implantation. The baseline and final LVEF were also higher in women. In HFmrEF, men had a higher prevalence of ischemic CMP, comorbidities, PCI, CABG, and baseline and final LVEF. Women had a higher prevalence of idiopathic and hypertensive CMP. In HFpEF, women were older and had a higher prevalence of hypertensive CMP and valve disease, comorbidities, valve surgery, and hospitalizations. Women also had a higher baseline and final LVEF. Men had a higher prevalence of ischemic CMP, MI, CKD, PCI, CABG, pacemaker, and ICD implantation. Men also had a higher baseline and final LVDD. In the HFrEF women and men, final LVEF increased, and LVDD decreased compared with respective baseline values. In HFmrEF women, final LVEF and LVDD increased compared with respective baseline values. In HFpEF women and men, final LVEF decreased, and LVDD increased compared with respective baseline values. During the follow-up period, death was higher in HFrEF men (26.4%) than in women (20.7%) (*p* < 0.0001). The cumulative incidence of death was higher in men than women in HFrEF but similar in other HF subtypes (Figure 2). In women, the variable comparisons of the different HF subtypes showed statistically significant differences for most of the variables analyzed, except for diabetes, stroke, CABG, and PCI (Table 3). Ischemic CMP and MI were more prevalent in HFmrEF than other HF subtypes. Idiopathic, hypertensive CMP and valve disease, CKD, AF, number of comorbidities, valve surgery, and pacemaker implantation were more prevalent, in descending order, in HFrEF and HFmrEF than HFpEF. ICD implantation, CRT, and heart transplants were more frequent in HFrEF than in other HF subtypes. The hospitalizations were higher in women with HFrEF and HFpEF. Mortality was higher in decreasing order in HFrEF and HFmrEF compared to HFpEF.

In men, the comparison of variables for the different HF subtypes showed statistically significant differences for most of the variables analyzed, except for CKD and stroke. Ischemic CMP and diabetes were more prevalent in HFmrEF and HFpEF than in HFrEF. MI was more prevalent in HFrEF and HFmrEF than in HFpEF. Idiopathic, the number of comorbidities, ICD, and CRT implantation were more prevalent, in descending order, in HFrEF, HFmrEF, and HFpEF. Hypertensive CMP and valve disease, AF, valve surgery, and pacemaker implantation were more prevalent in HFrEF and HFmrEF than in HFpEF. Heart transplants occurred more in HFrEF than in other HF subtypes. The hospitalizations were higher in women with HFrEF and HFpEF. Mortality was higher in HFrEF but similar in HFmrEF and HFpEF. 

Multivariable regression analysis for all patients adjusted for several covariates showed CKD, stroke, diabetes, AF, age, baseline LVEF, valve disease, MI, hypertensive CMP, and PCI + CABG but not sex as the independent predictors of all-cause death (Table 4).

Multivariable regression analysis for HF subtypes, adjusted for several covariates, showed CKD, stroke, diabetes, AF, hypertensive CMP, baseline LVEF, age, MI, valve disease, and PCI + CABG as the independent predictors of death for HFrEF. CKD, diabetes, stroke, AF, MI, age, valve disease, and PCI + CABG were independent predictors of death for HFmrEF. CKD, AF, age, diabetes, stroke, valve disease, and MI were independent predictors of death for HFpEF. Sex was not an independent predictor of death in all HF subtypes (Table 5).

## 4. Discussion

Our study showed a better prognosis in women with HFrEF, but after multivariate analysis adjusted for several covariates, sex was not an independent death predictor in all HF subtypes. Women with HFrEF had a lower prevalence of ischemic CMP and comorbidities. The LVEF was higher, and LVDD was lower in women than men. Our results were similar to those observed in large prospective cohort studies [6,13]. Gerber et al. showed higher but discrete cardiovascular mortality in men in a study conducted to estimate the HF incidence rates and outcomes after HF diagnosis between 2000 and 2010 in Olmsted County, Minnesota [6]. HF subtypes definition used the cutoff value of 50% for HFpEF (≥50%) or HFrEF (<50%), and the authors did not include the HFmrEF subtype but distributed it in HFrEF and HFpEF subtypes. The mortality was higher in HFrEF than in HFpEF and increased with aging. Men also had higher cardiovascular death that could be related to a more significant reduction in the incidence of HFrEF in women during the study period. Nevertheless, all-cause mortality was similar between women and men. In data analysis of two large cohort studies, Pandey et al. also showed almost two times higher lifetime risk of HFrEF (EF < 45%) in men compared to women aged >40 years but a similar lifetime risk for HFpEF (EF ≥ 45%) [13]. Patients with previous MI had a significantly increased lifetime risk for all HF subtypes. The lifetime risk increased by 2.5-fold for the incidence of HFpEF and 4-fold for HFrEF in the presence of previous MI. Men had a higher MI prevalence than women for all ten-year age groups, from 45 years old and for patients over 75 years. Patients with ischemic CMP had a higher death rate than other HF etiologies [19]. Likewise, mortality from ventricular arrhythmias in ischemic cardiomyopathy is higher in men than in women. HF associated with ventricular arrhythmia, which is more common in men, is associated with a worse prognosis and higher mortality [20,21]. These findings may explain the higher risk of death in men with HFrEF. Estrogen modulating various pathophysiologic processes in women may also partly explain why women have a better prognosis than men in HF [22,23]. Estrogen exerts a cardioprotective effect in HF by inhibiting sympathetic activity and the renin-angiotensin-aldosterone system. Estrogen decreases renin levels, angiotensin-converting enzyme activity, AT1 receptor density, and aldosterone production, and increases AT2 receptor density. Estrogen increases natriuretic peptides that intensify the inhibition of the renin-angiotensin-aldosterone system. Estrogen promotes better endothelial response to injury, prevents left ventricular remodeling and diastolic function, and protects the myocardial microvasculature. Estrogens’ protection at the cellular level is primarily carried out through increasing anti-oxidative defenses and maintaining mitochondrial integrity [24]. It is known that endogenous estrogens attenuate the inflammatory response and the expression of circulating MMP, which is directly associated with more significant ventricular dysfunction.

Our study also showed a higher prevalence of comorbidities in men for all HF subtypes, namely MI, CKD, and AF. These results agreed with the literature that showed a higher prevalence of comorbidities in men with HFrEF, while women have a higher prevalence of hypertension and diabetes but better renal function [16,25]. Women in all HF subtypes of our study had a higher prevalence of idiopathic and hypertensive CMP, similar to those observed in previous studies [26,27]. Our study also showed a higher prevalence of HFpEF in women. They were older and had a higher prevalence of hypertensive MCP and valvular disease, but in multivariate analysis, death in HFpEF patients was similar in women and men. Our results are similar to those in the literature for almost all clinical characteristics, echocardiographic data, and HF mortality [28,29,30]. Lam et al. showed a higher CV and all-cause death in women, but the risk of all-cause events was similar in women and men after adjusting to AF and renal dysfunction. On the other hand, Magnussen et al. showed higher mortality from HF in men than in women, even after adjusting for covariates that included the main risk factors for HF [30]. However, the population had baseline preserved renal function, and the authors did not include the renal function changes during the follow-up period in the multivariate analysis. It is well known that AF and CKD are comorbidities associated with higher mortality in women than men [31,32]. Our study showed a higher prevalence of comorbidities in women with HFpEF, including AF, compared to HFrEF and HFrEF. A high number of comorbidities in HFpEF could justify the same mortality observed in women and men. The worse prognosis in women with AF may also be related to inadequate treatment, such as less use of anticoagulants and fewer ablation procedures [33]. 

In our study, HFmrEF and HFpEF patients showed similar mortality in women and men. These findings differed from those in the literature, where mortality in patients with HFmrEF is intermediate between HFrEF and HFpEF [14,34,35,36,37]. The same mortality observed in patients with HFmrEF and HFpEF may be associated with a more significant number of comorbidities, such as MI and CKD, in HFmrEF patients. On the other hand, as observed in some studies, our study also showed that HFmrEF patients had similar characteristics to those with HFrEF and HFpEF [38,39]. In the HFmrEF group, patients were younger and had a higher prevalence of ischemic and hypertensive CMP and MI, which resembled those of the HFrEF group. The HFmrEF group was similar to HFpEF for hypertensive CMP, diabetes, and AF prevalence.

Our study has some limitations. First, it is a retrospective study in a single specialized tertiary care center where selection biases may occur and which includes patients with a more complex clinical picture. Second, an adequate definition of symptoms must be included, especially the NYHA functional class of dyspnea and other variables associated with a worse prognosis (ventricular arrhythmia, 6-min walk test). Third, we were unable to detail the cause of death adequately. Therefore, our analysis included cardiac and non-cardiac causes, including the deaths from COVID-19 which occurred between the pandemic months of March to September 2020. Finally, adequate information regarding drug treatment and dosages is needed.

## 5. Conclusions

Our large cohort study with many patients showed that women with HFrEF had a better prognosis than men. Nevertheless, after adjusting for several covariates in the multivariable analysis, death was similar in all women and men patients and also in women and men in the HF subtypes. Contrary to most previous studies, this study showed that sex was not an independent variable of all-cause mortality in HF subtypes. Our more recent data may result from improving the guideline-based therapy on women, reducing the previously existing therapeutic gap between women and men. CKD, stroke, diabetes, AF, MI, and age are the most critical variables of worse prognosis shared by all HF subtypes. Therefore, intensifying primary prevention of these comorbidities is essential to reduce HF morbidity and mortality. Future prospective cohort studies are needed to establish the actual HF subtypes prognosis in women and men.

## Figures and Tables

**Figure 1 ijerph-19-16171-f001:**
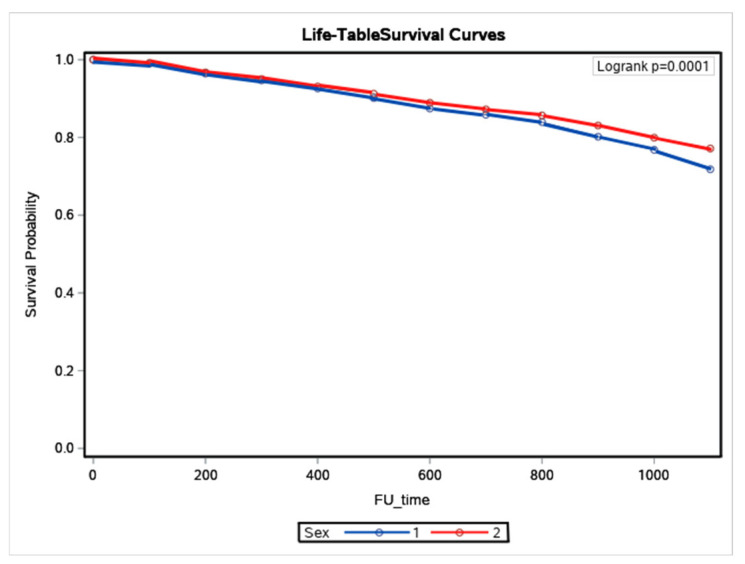
Kaplan–Meier survival curves in women and men with heart failure (Sex 1 means men and 2 women).

**Figure 2 ijerph-19-16171-f002:**
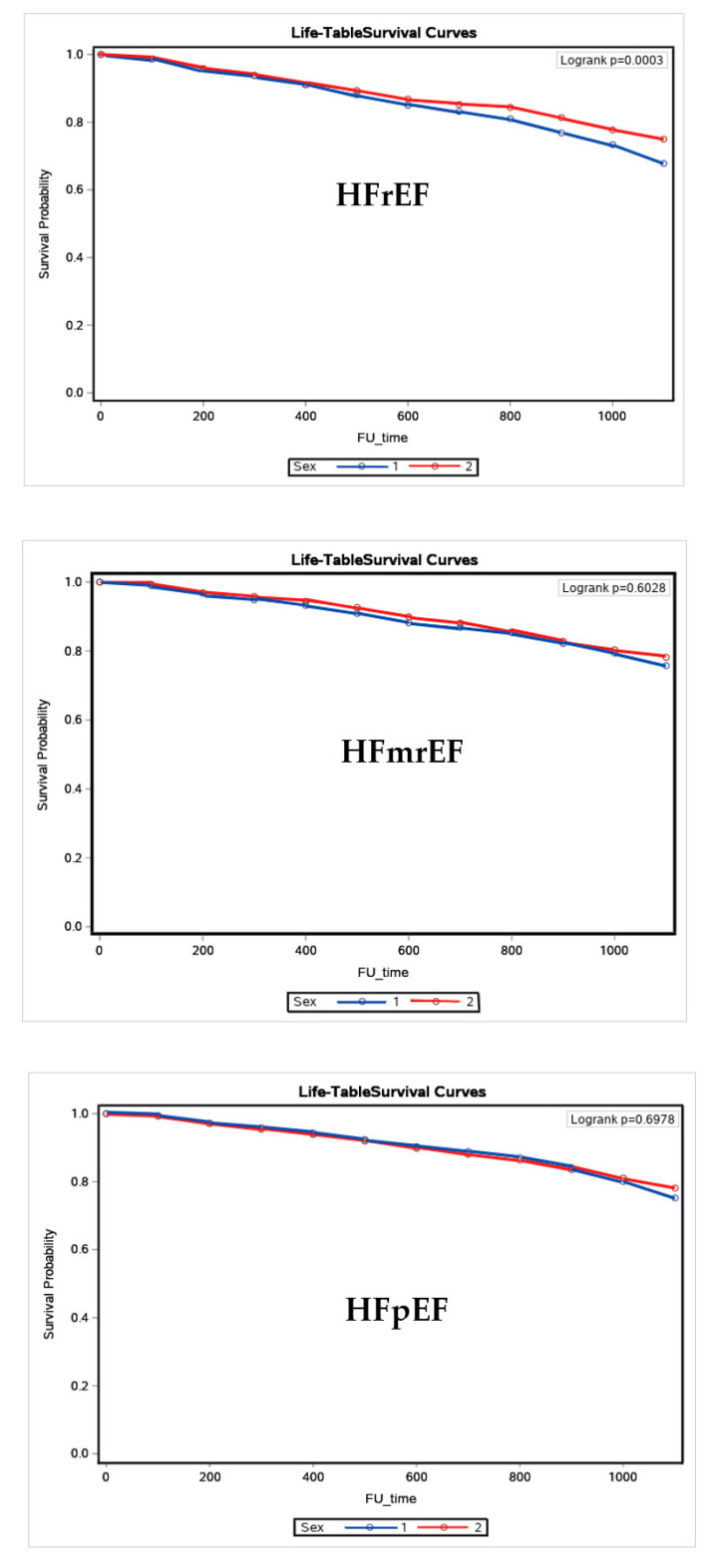
Kaplan–Meier survival curves in women and men with heart failure with reduced (HFrEF), mildly reduced (HFmrEF), and preserved (HFpEF) ejection fraction (Sex 1 means men and 2 women).

**Table 1 ijerph-19-16171-t001:** Clinical characteristics and echocardiographic data of the women and men with heart failure.

	All Patients N = 11,282	Women N = 5026 (44.6)	Men N = 6256 (55.4)	*p*
Age (Years)	63.9 ± 14.4	65.1 ± 15.0	63.0 ± 13.8	<0.0001
Ischemic CMP	3248 (28.8)	1126 (22.4)	2122 (33.9)	<0.0001
Idiopathic CMP	4598 (4S0.8)	1989 (39.6)	2609 (41.7)	<0.0022
Hypertensive CMP	1934 (17.1)	1033 (20.6)	901 (14.4)	<0.0001
Valve disease	1502 (13.3)	878 (17.5)	624 (10.0)	<0.0001
Myocardial infarction	1670 (14.8)	557 (11.1)	1113 (17.8)	<0.0001
Diabetes	214 (19.0)	934 (18.6)	1207 (19.3)	0.427
CKD	1202 (10.7)	416 (8.3)	786 (12.6)	<0.0001
Atrial fibrillation	2241 (19.9)	948 (18.9)	1293 (20.7)	0.019
Stroke	512 (4.5)	196 (3.9)	316 (5.1)	0.010
Number of comorbidities	5378 (47.7)	2225 (44.3)	3153 (50.4)	<0.0001
N = 1	3372 (29.9)	1482 (29.5)	1890 (30.2)	0.476
N = 2	1463 (13.0)	556 (11.1)	907 (14.5)	<0.0001
N = 3	446 (3.9)	155 (3.1)	291 (4.7)	0.149
N ≥ 4	97 (0.8)	32 (0.6)	65 (1.0)	0.489
PCI	397 (3.5)	140 (2.8)	257 (4.1)	<0.0001
CABG	884 (7.8)	285 (5.7)	599 (9.6)	<0.0001
Valve replacement	988 (8.8)	534 (10.6)	454 (7.3)	<0.0001
Valvoplasty	125 (1.1)	81 (1.6)	44 (0.7)	<0.0001
Pacemaker	622 (5.5)	339 (6.7)	283 (4.5)	<0.0001
ICD	265 (2.4)	72 (1.4)	193 (3.1)	<0.0001
CRT	317 (2.8)	151 (3.0)	166 (2.7)	0.181
Transplantation	176 (1.6)	62 (1.2)	114 (1.8)	0.007
Hospitalization	3207 (28.4)	1474 (29.3)	1733 (27.7)	<0.0001
LVEF baseline	45.3 ± 15.9	49.2 ± 16.0	42.3 ± 15.1	<0.0001
LVEF final	46.4 ± 15.0	49.9 ± 14.5 *	43.7 ± 14.7 *	<0.0001
LVDD baseline	57.7 ± 9.6	54.4 ± 8.8	60.1 ± 9.5	<0.0001
LVDD final	56.8 ± 10.0	53.7 ± 9.0 *	59.3 ± 10.0 *	<0.0001
HFrEF	4310 (38.2)	1488 (29.6)	2822 (45.1)	<0.0001
HFmrEF	1826 (16.2)	653 (13.0)	1173 (18.8)	<0.0001
HFpEF	5146 (45.6)	2885 (57.4)	2261 (36.1)	<0.0001
Death	2300 (20.4)	925 (18.4)	1375 (22.0)	<0.0001

* *p* < 0.05 for LVEF and LVDD baseline vs. final intragroup comparisons. Numbers in brackets mean percentage; *p* values: women vs. men; CABG: coronary artery bypass graft; CKD: chronic kidney disease; CMP: cardiomyopathy; CRT: cardiac resynchronization therapy; HFmrEF: HF with mildly reduced ejection fraction; HFpEF: HF with preserved ejection fraction; HFrEF: HF with reduced ejection fraction; ICD: implantable cardioverter-defibrillator; LVDD: left ventricular diastolic diameter; LVEF: left ventricular ejection fraction; PCI: percutaneous coronary intervention.

**Table 2 ijerph-19-16171-t002:** Clinical characteristics of the patients with heart failure with preserved (HFpEF), mid-range (HFmrEF), and reduced (HFrEF) left ventricular ejection fraction.

	HFrEF N = 4310 (38.2)	HFmrEF N = 1826 (16.2)	HFpEF N = 5146 (45.6)	*p*
Age (Years)	60.2 ± 13.7	64.1 ± 13.5	66.9 ± 14.6	<0.0001
Female	1488 (34.5)	653 (35.8)	2885 (56.1)	<0.0001
Cardiomyopathy				
Ischemic	1173 (27.2)	632 (34.6)	1443 (28.0)	<0.0001
Idiopathic	2353 (54.6)	754 (41.3)	1491 (29.0)	<0.0001
Hypertensive	616 (14.3)	314 (17.2)	1004 (19.5)	<0.0001
Valve disease	168 (3.9)	126 (6.9)	1208 (23.5)	<0.0001
Myocardial infarction	667 (15.5)	359 (19.7)	644 (12.5)	<0.0001
Diabetes	772 (17.9)	364 (19.9)	1005 (19.5)	0.044
CKD	544 (12.6)	212 (11.6)	446 (8.7)	<0.0001
Atrial fibrillation	670 (15.6)	363 (19.9)	1208 (23.5)	<0.0001
Stroke	217 (5.0)	80 (4.4)	215 (4.2)	0.129
Number of comorbidities	1931 (44.8)	915 (50.0)	2532 (49.2)	<0.0001
N = 1	1167 (27.1)	538 (29.5)	1667 (32.4)	0.153
N = 2	547 (12.7)	272 (14.9)	644 (12.5)	<0.0001
N = 3	180 (4.2)	82 (4.5)	184 (3.6)	0.157
N ≥ 4	37 (0.9)	23 (1.2)	37 (0.7)	0.118
PCI	114 (2.7)	76 (4.2)	207 (4.0)	0.0004
CABG	271 (6.3)	173 (9.5)	440 (8.6)	<0.0001
Valve replacement	119 (2.8)	87 (4.8)	782 (15.2)	<0.0001
Valvoplasty	6 (0.1)	8 (0.4)	111 (2.7)	<0.0001
Pacemaker	129 (3.0)	99 (5.4)	394 (7.7)	<0.0001
ICD	178 (4.1)	39 (2.1)	48 (0.9)	<0.0001
CRT	258 (6.0)	30 (1.6)	29 (0.6)	<0.0001
Transplantation	126 (2.9)	7 (0.4)	43 (0.8)	<0.0001
Hospitalization	1148 (26.6)	452 (24.7)	1607 (31.2)	<0.0001
LVEF baseline	28.7 ± 6.2	43.6 ± 2.9	61.3 ± 6.3	<0.0001
LVEF final	36.5 ± 12.9 *	44.2 ± 11.3	56.5 ± 11.1 *	<0.0001
LVDD baseline	64.8 ± 8.4	57.6 ± 6.5	51.0 ± 6.6	<0.0001
LVDD final	62.7 ± 10.4 *	58.0 ± 7.8 *	51.8 ± 7.4 *	<0.0001
Death	1054 (24.5)	342 (18.7)	904 (17.6)	<0.0001

* *p* values for comparisons between baseline and final means in the same group; CABG: coronary artery bypass graft; CKD: chronic kidney disease; CMP: cardiomyopathy; CRT: cardiac resynchronization therapy; HFmrEF: HF with mildly reduced ejection fraction; HFpEF: HF with preserved ejection fraction; HFrEF: HF with reduced ejection fraction; ICD: implantable cardioverter-defibrillator; LVDD: left ventricular diastolic diameter; LVEF: left ventricular ejection fraction; PCI: percutaneous coronary intervention.

**Table 3 ijerph-19-16171-t003:** Clinical characteristics of women and men with heart failure with preserved (HFpEF), mid-range (HFmrEF), and reduced left ventricular ejection fraction (HFrEF).

	HFrEF N = 4310 (38.2)		HFmrEF N = 1826 (16.2)		HFpEF N = 5146 (45.6)	
	Women N = 1488 (34.5)	Men N = 2822 (65.5) *	Women N = 653 (35.8)	Men N = 1173 (64.2) *	Women N = 2885 (56.1)	Men N = 2261 (43.9) *
Age (Years)	60.7 ± 14.1	60.0 ± 13.5	64.0 ± 14.1	64.1 ± 13.2	67.5 ± 15.1	66.2 ± 13.9 *
Cardiomyopathy						
Ischemic	320 (21.5)	853 (30.2) *	174 (26.7)	458 (39.1) *	632 (21.9)	811 (35.9) *
Idiopathic	847 (56.9)	1506 (53.4) *	295 (45.2)	459 (39.1) *	847 (29.4)	644 (28.5)
Hypertensive	264 (17.7)	352 (12.5) *	130 (19.9)	184 (15.7) *	639 (22.2)	365 (16.1) *
Valve disease	57 (3.8)	111 (3.9)	54 (8.3)	72 (6.1)	767 (26.6)	441 (19.5) *
Myocardial infarction	174 (11.7)	493 (17.5) *	97 (14.8)	262 (22.3) *	286 (9.9)	358 (15.6) *
Diabetes	274 (18.4)	498 (17.7)	199 (18.2)	245 (20.9)	541 (18.8)	464 (20.5)
CKD	162 (10.9)	382 (13.5) *	67 (10.3)	145 (12.4)	187 (6.5)	259 (11.5) *
Atrial fibrillation	166 (11.2)	504 (17.9) *	108 (16.5)	255 (21.7) *	674 (23.4)	534 (23.6)
Stroke	62 (4.2)	155 (5.5)	23 (3.5)	57 (4.9)	111 (3.9)	104 (4.6)
Number of comorbidities	588 (19.5)	1343 (57.3) *	288 (39.8)	627 (50.2) *	1349 (46.8)	1183 (44.5) *
N = 1	372 (25.0)	795 (28.2)	181 (27.7)	357 (30.4)	929 (32.2)	738 (32.6)
N = 2	155 (10.4)	392 (13.9)	79 (12.1)	193 (16.5)	322 (11.2)	322 (14.2)
N = 3	53 (3.6)	127 (4.5)	20 (3.1)	62 (5.3)	82 (2.8)	102 (4.5)
N ≥ 4	8 (0.5)	29 (1.0)	8 (1.2)	15 (1.3)	16 (1.6)	21 (0.9)
PCI	36 (2.4)	78 (2.8)	19 (2.9)	57 (4.9) *	85 (3.0)	122 (5.4) *
CABG	72 (4.8)	199 (7.1) *	47 (7.2)	126 (10.7) *	166 (5.8)	274 (12.1) *
Valve replacement	35 (2.4)	84 (3.0)	35 (5.4)	52 (4.4)	464 (16.1)	318 (14.1) *
Valvoplasty	1 (0.1)	5 (0.2)	4 (0.6)	4 (0.3)	76 (2.6)	35 (1.6) *
Pacemaker	46 (3.1)	83 (2.9)	39 (6.0)	60 (5.1)	254 (8.8)	140 (6.2) *
ICD	46 (3.1)	132 (4.7) *	12 (1.8)	27 (2.3)	14 (0.5)	34 (1.5) *
CRT	124 (8.3)	134 (4.8) *	13 (2.0)	17 (1.5)	14 (0.5)	15 (0.7)
Transplantation	43 (2.9)	83 (2.9)	1 (0.2)	6 (0.5)	18 (0.6)	25 (1.1)
Hospitalization	381 (25.6)	626 (22.2)	152 (23.3)	231 (19.7)	941 (32.6)	528 (23.9) *
LVEF baseline	29.2 ± 5.9	28.5 ± 6.3 *	43.7 ± 3.0	43.6 ± 2.9	62.2 ± 6.1	60.1 ± 6.3 *
LVEF final	38.0 ± 12.9	35.7 ± 12.8 *	44.7 ± 11.0	43.9 ± 11.5	58.2 ± 10.1	54.3 ± 11.8 *
LVDD baseline	62.6 ± 7.8	66.0 ± 8.4 *	56.0 ± 6.1	58.4 ± 6.5 *	49.4 ± 5.8	53.1 ± 6.8 *
LVDD final	60.3 ± 9.8	64.1 ± 10.4 *	56.6 ± 7.5	58.8 ± 7.9 *	50.0 ± 6.6	54.2 ± 7.6 *
Death	308 (20.7)	746 (26.4) *	120 (18.4)	222 (18.9)	497 (17.2)	407 (18.0)

Numbers in parenthesis mean percentage; * *p* < 0.05: comparisons between women vs. men in the same HF subtype. CABG: coronary artery bypass graft; CKD: chronic kidney disease; CMP: cardiomyopathy; CRT: cardiac resynchronization therapy; HFmrEF: heart failure with mildly reduced ejection fraction; HFpEF: heart failure with preserved ejection fraction; HFrEF: heart failure with reduced ejection fraction; ICD: implantable cardioverter-defibrillator; LVDD: left ventricular diastolic diameter; LVEF: left ventricular ejection fraction; PCI: percutaneous coronary intervention.

**Table 4 ijerph-19-16171-t004:** Cox regression analysis for death from heart failure adjusted for age, sex, ischemic cardiomyopathy, idiopathic cardiomyopathy, hypertensive cardiomyopathy (HCMP), valve disease, myocardial infarction (MI), diabetes, stroke, chronic kidney disease (CKD), atrial fibrillation (AF), baseline left ventricular ejection fraction (LVEF), percutaneous coronary intervention (PCI) + coronary artery bypass graft (CABG), and pacemaker + cardiac resynchronization therapy + implantable cardioverter-defibrillator.

	Hazard Ratio	95% Confidence Limits			Score Chi-Square	*p*
Age	1.02	1.01	1.02	CKD	1302	<0.0001
HCMP	0.81	0.72	0.92	Stroke	340	<0.0001
Valve disease	1.65	1.45	1.87	Diabetes	259	<0.0001
MI	1.29	1.16	1.43	AF	244	<0.0001
Diabetes	2.06	1.88	2.26	Age	72	<0.0001
Stroke	2.49	2.20	2.81	Valve disease	66	<0.0001
CKD	2.76	2.51	3.03	LVEF baseline	57	<0.0001
AF	1.87	1.71	2.04	MI	37	<0.0001
LVEF baseline	0.76	0.72	0.81	HCMP	11	0.0009
PCI + CABG	1.24	1.02	1.50	PCI + CABG	5	0.027

**Table 5 ijerph-19-16171-t005:** Cox regression analysis for death from heart failure of preserved (HFpEF), mid-range (HFmrEF), and reduced (HFrEF) left ventricular ejection fraction adjusted for age, sex, ischemic cardiomyopathy, idiopathic cardiomyopathy, hypertensive cardiomyopathy (HCMP), valve disease, myocardial infarction (MI), diabetes, stroke, chronic kidney disease (CKD), atrial fibrillation (AF), baseline left ventricular ejection fraction (LVEF), percutaneous coronary intervention (PCI) + coronary artery bypass graft (CABG), and pacemaker + cardiac resynchronization therapy + implantable cardioverter-defibrillator.

		Hazard Ratio	95% Confidence Limits			Score Chi-Square	*p*
HFrEF	Age	1.01	1.01	1.02	CKD	489	<0.0001
	HCMP	0.69	0.57	0.84	Stroke	176	<0.0001
	Valve disease	1.51	1.15	1.98	Diabetes	128	<0.0001
	MI	1.28	1.10	1.49	AF	78	<0.0001
	Diabetes	2.08	1.82	2.38	HCMP	24	<0.0001
	Stroke	2.65	2.21	3.18	Age	23	<0.0001
	CKD	2.70	2.35	3.09	LVEF baseline	22	<0.0001
	AF	1.80	1.57	2.06	MI	7	0.0049
	LVEF baseline	0.97	0.96	0.98	Valve disease	8	0.0031
	PCI + CABG	1.24	1.02	1.50	PCI + CABG	5	0.027
HFmrEF	Age	1.02	1.01	1.03	CKD	245	<0.0001
	Valve disease	2.00	1.40	2.86	Diabetes	65	<0.0001
	MI	1.78	1.39	2.27	Stroke	48	<0.0001
	Diabetes	2.33	1.86	2.93	AF	40	<0.0001
	Stroke	2.48	1.82	3.38	MI	20	<0.0001
	CKD	2.89	2.28	3.68	Age	14	0.0002
	AF	1.94	1.53	2.45	Valve disease	15	0.0001
	PCI + CABG	1.50	1.13	2.60	PCI + CABG	6	0.027
HFpEF	Age	1.03	1.03	1.04	CKD	565	<0.0001
	Valve disease	1.88	1.60	2.20	AF	161	<0.0001
	MI	1.22	1.01	1.46	Age	120	<0.0001
	Diabetes	2.08	1.80	2.40	Diabetes	79	<0.0001
	Stroke	2.14	1.76	2.61	Stroke	64	<0.0001
	CKD	2.87	2.46	3.35	Valve disease	57	<0.0001
	AF	1.94	1.69	2.23	MI	4	0.0344

## Data Availability

Not applicable.

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
