# Peer review of "Sex Differences in Heart Failure Mortality with Preserved, Mildly Reduced and Reduced Ejection Fraction: A Retrospective, Single-Center, Large-Cohort Study"

_ijerph, 2022, doi:10.3390/ijerph192316171_

Round 1
Reviewer 1 Report
The Authors decided to analyze the topic of gender differences in HF subtypes. This is a valuable approach; however I would like to underline major flaws of this study.
I suggest that Authors could change the language into more professional by using passive voice i.e. please do not use the form “we analyzed”, and change it into “… was analyzed” etc.
Please use the term “gender” instead of “sex”. The are tense mistakes. Please conduct extensive English corrections.
Introduction needs extension, please elaborate on the topic. Please add information regarding HF pathophysiology, symptoms, more detailed possible reasons of differences between women and men.
Line 43 The incidence of HF increases markedly after 60 years of age but more significantly in men – please provide citation.
Line 51 – Please be consistent and use HF abbreviation throughout whole manuscript.
Line 76 – “some” – Please provide number data.
Line 89 – Was the normality of the data checked?
Table 1. Please provide also the comparison regarding percentage of each condition to check the proportional differences in each subgroup.
Table 3. What are the numbers in brackets? SD? Percentage? Please explain.
Table 3. The group of women was significantly lower than men in total (5026 vs 6256). The study group should include similar (perfectly equal) number of women and men to provide reliable comparison between these groups in this context.
Please outline why this study is new/innovative comparing to other ones upon this topic.
Please also add more detailed possible reasons of differences between women and men in discussion section.
The reference list is way too short, please extend it.
Author Response
Reviewer 1.
Thanks for your time in reviewing our manuscript. The English language was edited. Below are the responses to the reviewer’s suggestions.
Comments and Suggestions for Authors
The Authors decided to analyze the topic of gender differences in HF subtypes. This is a valuable approach; however I would like to underline major flaws of this study.
I suggest that Authors could change the language into more professional by using passive voice i.e. please do not use the form “we analyzed”, and change it into “… was analyzed” etc.
Please use the term “gender” instead of “sex”. The are tense mistakes. Please conduct extensive English corrections.
RE: We think the term sex is appropriate for our population. The English language was extensively edited.
Introduction needs extension, please elaborate on the topic. Please add information regarding HF pathophysiology, symptoms, more detailed possible reasons of differences between women and men.
RE: OK, done.
Line 43 The incidence of HF increases markedly after 60 years of age but more significantly in men – please provide citation.
RE: OK. Citation included.
Line 51 – Please be consistent and use HF abbreviation throughout whole manuscript.
RE: OK, done.
Line 76 – “some” – Please provide number data.
RE: the exact number of Covid-19 deaths was unavailable.
Line 89 – Was the normality of the data checked?
RE: Normality was tested using the Kolmogorov-Smirnov test.
Table 1. Please provide also the comparison regarding percentage of each condition to check the proportional differences in each subgroup.
RE: OK. Done.
Table 3. What are the numbers in brackets? SD? Percentage? Please explain.
RE: OK. Mean percentage. The information is included in the footnotes.
Table 3. The group of women was significantly lower than men in total (5026 vs 6256). The study group should include similar (perfectly equal) number of women and men to provide reliable comparison between these groups in this context.
RE: This is a retrospective cohort study and not a case-control study. Statistical analysis methods used considered this difference.
Please outline why this study is new/innovative comparing to other ones upon this topic.
RE: OK. Text included in the Conclusions section.
Please also add more detailed possible reasons of differences between women and men in discussion section.
- Done.
The reference list is way too short, please extend it.
RE: OK. Done.
Reviewer 2 Report
First of all, I would like to thank you for the opportunity to review the article "Sex differences in mortality of congestive heart failure with preserved, mildly reduced, and reduced ejection fraction: A retrospective, single-center, large-cohort study". In this manuscript, the authors compare the prognosis (influence on the development of death from all causes) in patients with CHF separately in men and women. The authors obtained interesting results, showing that women had a better prognosis than men in HFrEF and similar mortality did for HFmrEF and HFpEF, but sex was not an independent predictor of death for all HF subtypes.
However, I have a number of questions and comments to which I would like answers from the authors.
1. It remains unclear from the manuscript what was the period of observation of patients.
2. Why age, sex, MI, diabetes, stroke, CKD, AF, PCI, CABG, and LVEF were chosen as covariates in the Cox proportional hazards model, although the groups differed in almost all indicators, in particular, in comorbidity, the frequency of valve disease, pacemaker, ICD.
3. When evaluating the dynamics of LVEF and LVDD, did the authors take into account the fact that 2300 patients died and they clearly did not undergo repeated ECHOCG examinations?
4. In section 2. Materials and Methods, the authors wrote, "The echocardiographic data included left ventricular ejection fraction (LVEF) and left ventricular systolic and diastolic diameters (LVDD) at baseline and the end of the study". Does this mean that all studies on surviving patients (nearly 9,000 patients) were conducted in September 2020?
5. What were the reasons for significant changes in LVEF and LVDD during the follow-up? The effect of drug therapy or the effect of bias due to a decrease in the number of examined patients?
6. In tables 4-5, two columns have the same name ("Variable"). Therefore, it is not clear which one the data in the other columns (eg Score Chi-Square and p) refer to. It needs to be explained.
Minor:
1. On page 3 (line 98) - the subheading "3. Results" should start on a new line and a new paragraph.
2. In table 1, there is no information in the notes about the meaning of the symbol *.
3. On page 5 (lines 141-143) in the phrase "The cumulative incidence of death was higher in HFrEF compared to HFmrEF (p<0.0001) and HFpEF (p<0.0001) but similar between HFmrEF and HFpEF (p=0.116) ( Figure 2)" contains an inaccuracy: Figure 2 refers to other data (namely, Kaplan-Meier survival curves in women and men with heart failure with reduced (HFrEF), mildly reduced (HFmrEF), and preserved (HFpEF) ejection fraction).
4. In my opinion, the sizes of the graphs in Figure 2 are small, it is better to increase them at least to the size of Figure 1.
Author Response
Reviewer 2.
Thanks for your time in reviewing our manuscript. The English language was edited. Below are the responses to the reviewer’s suggestions.
Comments and Suggestions for Authors
First of all, I would like to thank you for the opportunity to review the article "Sex differences in mortality of congestive heart failure with preserved, mildly reduced, and reduced ejection fraction: A retrospective, single-center, large-cohort study". In this manuscript, the authors compare the prognosis (influence on the development of death from all causes) in patients with CHF separately in men and women. The authors obtained interesting results, showing that women had a better prognosis than men in HFrEF and similar mortality did for HFmrEF and HFpEF, but sex was not an independent predictor of death for all HF subtypes.
However, I have a number of questions and comments to which I would like answers from the authors.
- It remains unclear from the manuscript what was the period of observation of patients.
RE: As stated in the Methods section, the follow-up period was from February 2017 to September 2020.
- Why age, sex, MI, diabetes, stroke, CKD, AF, PCI, CABG, and LVEF were chosen as covariates in the Cox proportional hazards model, although the groups differed in almost all indicators, in particular, in comorbidity, the frequency of valve disease, pacemaker, ICD.
RE: We also included in COX multivariate analysis the surgical variables grouped in PCI+CABG and PM+RCT+ICD as covariates. The previous results did not change for all patients, but coronary intervention (PCI+CABG) was an independent covariate for all-cause of death for HFrEF and HFmrEF.
- When evaluating the dynamics of LVEF and LVDD, did the authors take into account the fact that 2300 patients died and they clearly did not undergo repeated ECHOCG examinations?
RE: No, we did not consider this fact because the death occurred in close percentages between 18% to 22% in the HF subtypes, despite being statistically significant at univariate analysis, and we think the missing final echocardiograms may not have influenced the means LVEF and LDDD results in HF subtypes.
- In section 2. Materials and Methods, the authors wrote, "The echocardiographic data included left ventricular ejection fraction (LVEF) and left ventricular systolic and diastolic diameters (LVDD) at baseline and the end of the study". Does this mean that all studies on surviving patients (nearly 9,000 patients) were conducted in September 2020?
RE: No, the last echocardiogram closer to September 2020 was considered. We included this information in the Methods section.
- What were the reasons for significant changes in LVEF and LVDD during the follow-up? The effect of drug therapy or the effect of bias due to a decrease in the number of examined patients?
RE: These results were likely due to possible improvements in clinical and interventional management. As we are unsure, we did not detail these findings.
- In tables 4-5, two columns have the same name ("Variable"). Therefore, it is not clear which one the data in the other columns (eg Score Chi-Square and p) refer to. It needs to be explained.
RE: We removed the unnecessary word, but it only meant the column of variables cited below in the Table.
Minor:
- On page 3 (line 98) - the subheading "3. Results" should start on a new line and a new paragraph.
RE: OK. Done
- In table 1, there is no information in the notes about the meaning of the symbol *.
RE: OK. Done
- On page 5 (lines 141-143) in the phrase "The cumulative incidence of death was higher in HFrEF compared to HFmrEF (p<0.0001) and HFpEF (p<0.0001) but similar between HFmrEF and HFpEF (p=0.116) ( Figure 2)" contains an inaccuracy: Figure 2 refers to other data (namely, Kaplan-Meier survival curves in women and men with heart failure with reduced (HFrEF), mildly reduced (HFmrEF), and preserved (HFpEF) ejection fraction).
RE: OK. Corrected.
- In my opinion, the sizes of the graphs in Figure 2 are small, it is better to increase them at least to the size of Figure 1.
RE: OK. Resized the Figure 2.
Round 2
Reviewer 1 Report
I would like to thank the Authors for the replies to my review. The Authors replied to all my comments, however I am not fully satisfied with all the responses.
I believe that the list of relevant references should be longer.
Author Response
Reviewer 1
Comments and Suggestions for Authors
I would like to thank the Authors for the replies to my review. The Authors replied to all my comments, however I am not fully satisfied with all the responses.
I believe that the list of relevant references should be longer.
RE: Thanks to the reviewer for the time spent reviewing our article. We increased the number of references, but we accept further suggestions from the reviewer if any reference was missing.
Reviewer 2 Report
Thanks to the authors for the work done to correct the manuscript. I am not entirely satisfied with the authors' response to my 3rd remark. The authors state: "No, we did not consider this fact because the death occurred in close percentages between 18% to 22% in the HF subtypes, despite being statistically significant at univariate analysis, and we think the missing final echocardiograms may not have influence the means LVEF and LDDD results in HF subtypes". I believe that the answer is not entirely correct, since among the deceased patients there could be just patients, for example, with the lowest LVEF (group of patients with HFrEF, table 3). And due to this, there was an increase in LVEF in this group. I think that some incorrectness of such calculations should be indicated in the study limitations section. Or evaluate the dynamics of LVEF and LVDD only in those patients in whom both the initial and final results of ECHOCG are available
Author Response
Thanks to the authors for the work done to correct the manuscript. I am not entirely satisfied with the authors' response to my 3rd remark. The authors state: "No, we did not consider this fact because the death occurred in close percentages between 18% to 22% in the HF subtypes, despite being statistically significant at univariate analysis, and we think the missing final echocardiograms may not have influence the means LVEF and LDDD results in HF subtypes". I believe that the answer is not entirely correct, since among the deceased patients there could be just patients, for example, with the lowest LVEF (group of patients with HFrEF, table 3). And due to this, there was an increase in LVEF in this group. I think that some incorrectness of such calculations should be indicated in the study limitations section. Or evaluate the dynamics of LVEF and LVDD only in those patients in whom both the initial and final results of ECHOCG are available.
RE: Thanks to the reviewer for the time spent reviewing our article. We reanalyzed LVEF and LVDD data, including only patients with available baseline and follow-up echocardiograms, as suggested by the reviewer.